# Household dietary diversity across regions in Ethiopia: Evidence from Ethiopian socio-economic survey data

**Workicho Jateno**[1]*, **Bamlaku Alamirew Alemu**[2], **Maru Shete**[3]

1 College of Development Studies, Addis Ababa University, Addis Ababa, Ethiopia, 2 Yom Institute of Economic Development, Addis Ababa, Ethiopia, 3 Department of Development Economics and Management, Ethiopian Civil Service University, Addis Ababa, Ethiopia

* workicho.jateno@gmail.com

## Abstract

### Background

Household food and nutrition insecurity continued to be a development and policy agenda in Ethiopia. Assessing the patterns and determinants of household dietary diversity is an important area of research given its importance for policy uptake in the country. This study is, therefore, initiated to identify the dominant food groups consumed by households and to investigate the determinants of household dietary diversity in the country.

### Method

We used data from the 4th wave of the Ethiopian socioeconomic survey. The survey data for this study included 3,115 households living in rural areas (hereafter called 'rural households'). Household Dietary Diversity Score (HDDS) was calculated and categorized as per the FAO's recommendation: low HDDS category for those who consume three or less food groups, medium HDDS for those who consume four to six, and high HDDS for those who consume seven and more food groups during the past seven days. Ordinal logistic regression model was employed to estimate the determinants of rural household's dietary diversity.

### Results

Cereals were the most dominant food group consumed by 96.4% of the households followed by pulses, which was consumed by 82% of the households Nutrition-dense food commodities such as lean meat, vegetables and fruits were the least consumed food groups by households in Ethiopia. In terms of determinants of dietary diversity, female headed households have 38% more chance of consuming diverse foods compared to their male-headed counterparts (AOR = 1.38, 95% CI: 1.10, 1.73). Household heads who completed secondary education and above have 62% more chance of consuming diverse foods compared to uneducated household heads (AOR = 1.62, 95% CI: 1.2, 2.30). Household heads who are single have 37% less chance of consuming diverse foods compared to those household

**Data Availability Statement:** The data underlying the results presented in the study are available from World Bank and Central Statistical Agency: Socioeconomic Survey 2018-2019. Open Access

**Funding:** The author(s) received no specific funding for this work.

**Competing interests:** The authors have declared that no competing interests exist.

heads who are married (AOR = 0.63, 95% CI: 0.50, 0.80). Those households located in Harari regional state and in the rural surroundings of Diredawa town have 6.56 times more chance of consuming diverse foods compared to those households living in Tigray and Amhara regional states (AOR = 6.56, 95% CI: 4.60, 9.37). The results also highlighted that households who are in the upper wealth category have 9 times more chance of consuming diverse foods compared to those households who are the lower wealth category (AOR = 8.54, 95% CI: 6.79, 11.98).

## 1. Background of the study

The concept of household food security is a more recent development. The bulk of literature dated from 1980s equated national food security with food self-sufficiency. Food self-sufficiency is an essential but not a sufficient vehicle for solving household level malnutrition and food insecurity problems [1]. Food security is a concept that evolved over time. It is defined as the availability and access of food to all people at all times, as well as utilization and stability of foodstuff for a healthy life. More specifically, *food security is achieved when all people at all times have physical, social and economic access to sufficient, safe and nutritious food to meet their dietary needs and food preferences for an active and healthy* life [2]. Literature provides frameworks (e.g Sen's entitlement framework), approaches (e.g the Sustainable Livelihood Approach) and conceptual models (e.g UNICEF's model of malnutrition) to analyses and explain food and nutrition security situation of families. In this regard, [3] presented four entitlements namely, production-based entitlement, trade-based entitlement, own-labour entitlement, and inheritance and transfer entitlement. According to him, families endowed with these entitlements can escape from famine trap. Production-based entitlement improves food security situation of families through improving availability of food, and trade-based entitlement, own labour entitlement, and inheritance and transfer entitlement improve food security situation of families by creating access to food. Similarly, the Sustainable Livelihood Approach (SLA) presented by [4] gave a broader perspective of Sen's entitlements (known as natural, human, social, financial and physical capital). The five capitals presented in SLA also serve to explain food security situation of families. Households endowed with natural and physical capital will have the potential to produce more food, which increases food availability. Financial, social and human capitals improve the capacity of families to access food.

Nutrition security, on the other hand, emphasizes on the utilization dimension that focuses on the intake of a wide range of foods. Therefore, nutrition is an integral component of food security [5]. A conceptual model developed by [6] highlighted that resources and societal norms are important determinants of improved availability and utilization of foods, which in turn improves dietary diversity of families. Furthermore, in the conceptualization of food and nutrition security, there is a need to bring the two aspects together. The food security framework emphasizes on economic approach in which food is considered as a commodity while nutrition framework adopts a biological approach in which human beings are the starting point. Hence, in order to address improved household nutritional status, one has to deal with food security aspect, which is improving access for diversified food [7].

Achievement of optimum consumption of diversified foods is important in developing countries, such as Ethiopia, to contribute to the efforts of ending all forms of malnutrition [8, 9]. The study by [10] highlighted that dietary diversity correlates with household's per capita consumption implying that households that consume more diverse foods, they also have more access to food. Hence, consumption of variety of food groups is associated with greater energy

intake [11, 12]. Besides, household's dietary diversity is meant to provide an indication of household's economic access to food. An increase in dietary score means consumption of increased number of food groups [13]. Moreover, evidence show that household-level dietary diversity is strongly associated with household's per capita income and energy availability suggesting that dietary diversity could be a useful indicator of the access dimension of food security [14].

Ethiopia is one of the countries with the highest malnutrition problems. Thirty-eight percent of children under 5 are stunted, 10% are wasted, and 24% are underweight [15]. Ethiopia accounts for one of the countries with the large malnutrition burden in Africa. Based on the 2021 Global Hunger Index, Ethiopia is positioned 90[th] out of the 116 countries in terms of the rate of malnutrition. Furthermore, the burden of micronutrient deficiencies, notably iron, iodine, zinc and vitamin A, is among the highest in the country [16]. This is partly due to consumption of less diverse foods (i.e, low dietary diversity) as Ethiopian diets are particularly low in chicken, fruits, vegetables, and red meat [17].

There are a number of socioeconomic and demographic factors affecting dietary diversity of households. In Ethiopia, some studies have been conducted in order to analyze the determinants of household's dietary diversity [9, 18, 19]. Although these studies revealed the importance of household size, age, sex, level of education of household and land size as factors determining rural household's dietary diversity, they are case studies focusing on limited geographical scope. Undertaking research accounting for regional variations would help to pinpoint directions for national policymaking, as lifestyles in rural Ethiopia cannot be considered uniform. The policy actions that help achieve food and nutrition security in one context may not yield the same result elsewhere. Towards this end, there is a need to get a nuanced understanding as to whether regional differences correlates with dietary diversity with the implications of designing and implementing region-specific food and nutrition security programmes. This study is, therefore, designed with the objective of identifying the dominant food groups consumed among rural households in Ethiopia, and to investigate the determinants of household dietary diversity across regions in the country. The findings would contribute to evidence-based policy development, and craft relevant interventions by government and development partners.

## 2. Methods

### 2.1. Data source and variables

The data for this study came from the 2018/2019 Ethiopia Socioeconomic Survey (ESS) collected by the Central Statistics Agency of Ethiopia (CSA, now renamed as Ethiopian Statistical Services) in collaboration with the World Bank [20]. The ESS survey was conducted in 10 regions of the country. It included 7,527 households (both rural and urban) from 565 Enumeration Areas (EA). EA is smallest statistical sample unit. The households were selected through multi-stage sampling procedures. In the first stage, the EAs were selected using a simple random sampling procedure. In the second stage, the households to be surveyed from each of the EAs were selected using systematic random sampling. While the data set included households living in rural and urban areas, the scope of this study focused only on rural households in all regions of the country. Accordingly, only 3115 households residing in the rural parts of the country were included.

### 2.2. Variables

The concept of household food security is multidimensional by its nature. It, thus, require different approaches and indicators of measurement. Hoddinnott [10, 21] presented four

methods of measuring household food security: individual calorie intake, household caloric consumption, dietary diversity, and indices of household coping strategies. Household Dietary Diversity (HDD) is measured using the number of food groups consumed over a reference period. It reflects that consumption of variety of foods and food groups ensures adequate intake of essential nutrients [13]. In this study, 10 food groups were included to assess the dietary diversity of rural households in Ethiopia. A single point was given to each of the food groups consumed over the past seven days giving a maximum sum of 10 points for the total dietary diversity score for each household. Following this, Household Dietary Diversity Score (HDDS) was developed for each respondent. According to [13], when households consume three or less food groups in the past seven days, they are categorized under low HDDS. Similarly, those households that consume four to six food groups and seven or more food groups are categorized under medium HDDS and high HDDS category respectively. According to [22], HDDS measures the access component of household's food security. As the HDDS increases, the access to diversified food groups also increases. Commonly, food security studies in Ethiopia adopted individual calorie intake/ household caloric consumption, which focuses on the total caloric intake by households than dietary quality. In our study, since the focus is on dietary quality rather than on total energy intake, we used the HDDS that takes into account consumption of diverse foods. The indicator provide a more nuanced understanding of dietary quality than the calorie consumption method, which only considers total energy intake. The measurement indicator (i.e dietary diversity) is also scientifically supported by various studies [13, 22] as a suitable indicator of dietary quality. Therefore, this study adopted the HDDS method of measuring household's food security.

In this study, the independent variables included to explain household's dietary diversity where household size, sex of the household head, education level of the household head, marital status of the household head, religion of the household head, participation in the Productive Safety Net Program (PSNP), location of the household, and wealth index. As one of the explanatory variables, this study constructed household's wealth index using the Principal Component Analysis (PCA). In the construction of wealth index using the PCA household assets, such as livestock, type of house and house ownership, household assets and agricultural land ownership were included. In developing the index, first, the variables were coded, then entered into Stata software, and then analyzed using the PCA technique. The variables that have a communality value of greater than 0.5 were used to produce the factor scores. Finally, the factor scores were summed up to produce the index. After the wealth index was constructed, we further categorized the households into successive quantiles. The first quantile constituted the poorest households, the second quantile make up the poorer group of households, the third quantile constituted the middle wealth group, and the fourth and fifth quantiles constituted the rich and the richest households respectively.

### 2.3. Ethics statement

We have used the fourth round of a panel dataset produced at national level in Ethiopia, which is publicly accessible. The secondary data were produced by the Federal Democratic Republic of Ethiopia (through the Central Statistics Agency of Ethiopia) funded by the World Bank, Bill and Melinda Gates Foundation and the Foreign Commonwealth and Development Office (see: https://microdata.worldbank.org/index.php/catalog/3823#metadata-producers_sponsors) [20]. The production of the panel dataset was conducted with the applications of ethical elements such as participant's consents and confidentiality of personal information. Consents of participants were asked verbally. This can be objectively verified from the following link, which states that some participants were not willing to participate in the survey,

which reduced the response rate to 85% (refer to the section, 'Response rate' in the following link: https://microdata.worldbank.org/index.php/catalog/2783) [20]. Key confidentiality statements such as: (i) names of the respondents, (ii) village and constituency names, (iii) descriptions of household dwelling and agricultural field locations, (iv) phone numbers of household members and their reference contacts, (v) GPS-based dwelling and agricultural field locations, (vi) names of the children of the head/spouse living elsewhere, (vii) names of the deceased household members, (viii) names of individuals listed in the network roster, and (ix) names of field staff were not included (see: https://microdata.worldbank.org/index.php/catalog/3823#metadata-disclaimer_copyright) [20].

## 2.4. Method of data analysis

First, we downloaded the dataset for the study from the World Bank Microdata Library (see: https://microdata.worldbank.org/indefromx.php/catalog/3823) [20]. Then, we extracted the study variables, coded and recoded them, created new variables (Example, wealth index), worked on missing values and outliers. We used descriptive statistics such as mean, standard deviation, median, interquartile range and percentage to describe the data. We examined association/difference between variables using chi-square test, Fisher exact test, independent t-test, one-way analysis of variance (ANOVA), and the Wilcoxon signed rank test.

In examining the determinants of HDD, literature suggests three different analytical models. These are poison regression model by considering HDD as a count data [13, 23], ordered logit model by considering HDD as ordered values [24] and multinomial logit model by considering HDD as categorical but non-ordered values [13]. In this study, we adopted the ordered logit model to estimate the determinants of HDD since the dependent variable has an ordered nature–low, medium, and high dietary diversity. The ordered logit model estimates the underlying tendency of an observed phenomenon taking into account a vector of explanatory variables and a random error term [25]. Variance Inflation Factor (VIF) was employed to test multicollinearity problem among independent variables. Variables with VIF values less than 10 are considered to have no multicollinearity problem. Conversely, those with VIF values above 10 are considered to have problem of multicollinearity and should be excluded from the model. In this study, the VIF test results showed that there is no problem of multicollinearity. We have also checked the proportional odds assumption of the ordered logit model using the Brant test. The result showed that the model did not violate the proportional odds assumptions. All the analyses were conducted using STATA version 14.

## 3. Results and discussion

### 3.1. Household characteristics

The statistical evidence in Table 1 shows that the mean household size is 5 while the mean age is 44.4 years. About 32 percent of the age of the households falls above 50 years. Out of the households, 73.5 percent were male-headed while the remaining 26.5 percent were female-headed. In most cases a household headed by females are better in the consumption of diverse foods compared to male-headed households due to the cultural responsibility of females in Ethiopia in the preparation of foods. They have also better knowledge on variety of foods than males [26]. The result for the marital status of household heads indicated that the majority of the households are currently married (77 percent). Those households who follow Christianity as a religion are found to be more than 50 percent. The proportion of Muslims constituted 46.4 percent followed by Orthodox Christians who make up 36.6 percent. A study by [27] disclosed the importance of religion in the study of household diets among Ethiopian. During fasting seasons among Orthodox Christians in Ethiopia, vegetables are more consumed than

**Table 1. Description of household demographic and socioeconomic characteristics.**

| Characteristics | Frequency | Percent | Mean (Std. Dev) |
|---|---:|---:|---:|
| **Household head sex** | | | |
| Male | 2290 | 73.50 | |
| Female | 825 | 26.50 | |
| **Age of household head** | | | **44.4 (15.4)** |
| **Education level of household head** | | | |
| Illiterate | 2042 | 65.55 | |
| Primary | 833 | 26.74 | |
| Secondary and above | 240 | 7.70 | |
| **Marital status of household head** | | | |
| Currently married | 2396 | 76.97 | |
| Currently single | 717 | 23.03 | |
| **Religion of household head** | | | |
| Orthodox | 1138 | 36.57 | |
| Muslim | 1443 | 46.37 | |
| Protestant and others | 531 | 17.06 | |
| **Household size** | | | **5 (2.29)** |
| 1–3 | 969 | 31.10 | |
| 4–5 | 1029 | 33.00 | |
| 6 or more | 1117 | 35.90 | |
| **Regions** | | | |
| Tigray | 393 | 12.62 | |
| Afar | 299 | 9.60 | |
| Amhara | 479 | 15.38 | |
| Oromia | 453 | 14.54 | |
| Somali | 355 | 11.40 | |
| Benishangul Gumuz | 169 | 5.43 | |
| SNNP | 422 | 13.55 | |
| Gambela | 195 | 6.26 | |
| Harar | 190 | 6.10 | |
| Dire Dawa | 160 | 5.14 | |
| **Wealth Index** | | | |
| Poorest | 623 | 20.03 | 0.58 (0.52) |
| Poorer | 622 | 19.99 | 0.73 (0.55) |
| Middle | 622 | 19.99 | 0.82 (0.55) |
| Richer | 622 | 19.99 | 0.91(0.53) |
| Richest | 622 | 19.99 | 1.12(0.56) |
| **PSNP status** | | | |
| Yes | 2423 | 77.78 | |
| No | 692 | 22.22 | |

dairy products compared to period of non-fasting period [28, 29]. Anecdotal evidence also show that among the Muslim believers, consumption of milk and animal proteins are common during fasting periods. Education level of household head indicated that the majority of households (65.6 percent) are illiterate. Education is presumed to increase access to diverse foods from purchases through improved labour-based entitlement. It also affects the level of awareness of families on the benefits of consuming diverse foods. The wealth index analysis revealed there is significant difference among the wealth groups (F = 4.59; p<0.01). Those households

who fall under the poorest wealth category have 0.58 asset score while the richest households have an asset score of 1.12. Furthermore, it was found that the majority of households (77.8 percent) were not beneficiaries of PSNP.

## 3.2. Patterns of household's dietary diversity in Ethiopia

The overall analysis of dietary diversity of households indicated that about 91 percent of the households (65 percent under medium and 26 percent under low category) fall between low and medium dietary diversity category with only 9 percent of them falling under high dietary diversity category. This reveals the fact that the majority of the rural households in Ethiopia have limited dietary diversification. On average, the mean number of food groups consumed by rural households in Ethiopia was found to be five commodities with a dietary diversity ranging between two to ten food commodities.

Table 2 presents the results of chi-square analysis to examine differences in the status of household dietary diversity against the characteristics of the study respondents. The finding highlighted the existence of statistically significant differences in household dietary diversity across regions (P<0.01). Except for Harari and Diredawa, the majority of households fall under medium dietary diversity (more than 60 percent). In general, in these regions, those who fall under high dietary diversity are below 15 percent. In the case of Harari and Diredawa, 29 percent of them fall under high dietary diversity. The possible reason would be those households from these two regions reside in proximity to the towns, and hence, able to get access to diverse foods from markets. Second, they may have also better employment opportunities that increases their incomes that further improve their purchasing ability to access to diverse foods from the markets. Third, due to cultural differences compared to households in Amhara, Oromia and Tigray regional states, food habits of families in these areas may improve the propensity of consuming diverse foods.

The findings further showed that there are significant differences in household's dietary diversity that have different socio-economic and demographic characteristics. The analysis of age of households indicated that as age of heads of the households moves from the highest age bracket (i.e >50) to the lowest age bracket (i.e < = 30), the proportion of households who fall under high dietary diversity increases from 6.9 percent to 13.4 percent. This implies that households headed by younger heads have the tendency of consuming more diverse foods than households headed by older heads. In the case of education, those heads of households who completed primary education are proportionally better to fall under high dietary diversity category (26.67 percent) compared to the other two groups (illiterates who constituted 6.61 percent, and secondary school completed who constituted 9.25 percent). The finding further highlighted that the majority of households who did not participate in PSNP (64.8 percent) fall under medium dietary diversity category. Generally speaking, it was found that those households who did not participate in the PSNP fall under higher category compared to those who participated (Table 2).

The result in Fig 1 highlights the outcome of the analysis of different type of food groups consumed by households. Cereal was the most dominant food group consumed by 96.4 percent of the households. This is in line with the findings of [16, 19]. Pulses are the second dominant food group consumed by 82 percent of the households. On the other hand, consumption of meat and egg were least consumed food group among the households, which are also in line with the findings of [19]. Furthermore, households in Ethiopia do not dominantly consume nutrition-dense food commodities such as lean meat, vegetables and fruits. In developing countries like Ethiopia, dietary diversity is a challenge among rural people. In most cases, their diets are based on starchy staples with inadequate animal products, fresh fruits and vegetables.

**Table 2. Percentage distribution of household DD according to the characteristics of the study respondents and households, Ethiopia, 2018/19.**

| Characteristics | Low dietary diversity | | Medium dietary diversity | | High dietary diversity | | Chi² |
|---|---|---|---|---|---|---|---|
| | N | Percent | N | Percent | N | Percent | |
| **Sex of household head** | | | | | | | 8.87** |
| Male | 568 | 24.80 | 1501 | 65.60 | 221 | 9.70 | |
| Female | 247 | 29.90 | 511 | 61.90 | 67 | 8.10 | |
| Age of household head | 815 | 26.16 | 2012 | 64.59 | 288.0 | 9.25 | 29.48*** |
| < = 30 | 166 | 24.5 | 421 | 62.1 | 91 | 13.4 | |
| 31–40 | 190 | 23.3 | 549 | 67.3 | 77 | 9.4 | |
| 41–50 | 165 | 26 | 416 | 66 | 52 | 8 | |
| >50 | 294 | 29.7 | 628 | 63.4 | 68 | 6.9 | |
| **Education level of household head** | | | | | | | 99.62*** |
| Illiterate | 612 | 29.97 | 1295 | 63.42 | 135 | 6.61 | |
| Primary | 168 | 14.58 | 153 | 63.75 | 52 | 21.67 | |
| Secondary and above | 815 | 26.16 | 2012 | 64.59 | 288 | 9.25 | |
| **Marital status of household head** | | | | | | | 32.18*** |
| Currently married | 569 | 23.75 | 1591 | 66.40 | 236 | 9.85 | |
| Currently single | 245 | 34.17 | 420 | 58.58 | 52 | 5.25 | |
| **Religion of household head** | | | | | | | 25.16*** |
| Orthodox | 332 | 29.17 | 729 | 64.06 | 77 | 6.77 | |
| Muslim | 359 | 24.88 | 917 | 63.55 | 167 | 11.57 | |
| Protestant and others | 122 | 22.98 | 365 | 68.74 | 44 | 8.29 | |
| **Household size** | | | | | | | 20.24*** |
| 1–3 | 302 | 31.17 | 589 | 60.78 | 78 | 8.05 | |
| 4–5 | 249 | 24.20 | 672 | 65.31 | 108 | 10.50 | |
| 6 or more | 264 | 23.63 | 751 | 67.23 | 102.0 | 9.13 | |
| **Regions clustered** | | | | | | | 267.02*** |
| Tigray and Amhara | 301 | 34.52 | 535 | 61.35 | 36 | 4.13 | |
| Oromia and SNNP | 194 | 22.17 | 602 | 68.80 | 79 | 9.03 | |
| Benshangul Gumz and Gambela | 98 | 26.92 | 220 | 60.44 | 46 | 12.64 | |
| Somali and Afar | 183 | 27.98 | 446 | 68.20 | 25 | 3.82 | |
| Diredawa and Harare | 39 | 11.14 | 209 | 59.71 | 102 | 29.14 | |
| **Wealth Index** | | | | | | | 333.61*** |
| Poorest | 271 | 43.50 | 344 | 55.22 | 8 | 1.28 | |
| Poorer | 199 | 31.99 | 392 | 63.02 | 31 | 4.98 | |
| Middle | 163 | 26.21 | 411 | 66.08 | 62 | 9.97 | |
| Richer | 166 | 18.65 | 444 | 71.38 | 62 | 9.97 | |
| Richest | 65 | 10.45 | 420 | 67.52 | 137 | 22.03 | |
| **PSNP status** | | | | | | | 9.63** |
| Yes | 204 | 29.48 | 441 | 63.7 | 47 | 6.79 | |
| No | 611 | 25.22 | 1571 | 64.8 | 241 | 9.25 | |

*** Significant at p<0.01

**Significant at p<0.05

* Significant at p<0.1

Due to resource constraints, they lack access to adequate and diversified diets [30]. Our findings also highlighted similarity of food group consumption patterns between regions in Ethiopia. More specifically, similarities are observed between Amhara and Tigray; Somali and Afar; Oromai and SNNPR; Benshagul Gumze and Gambela; and Harari and Diredawa (Fig 1).

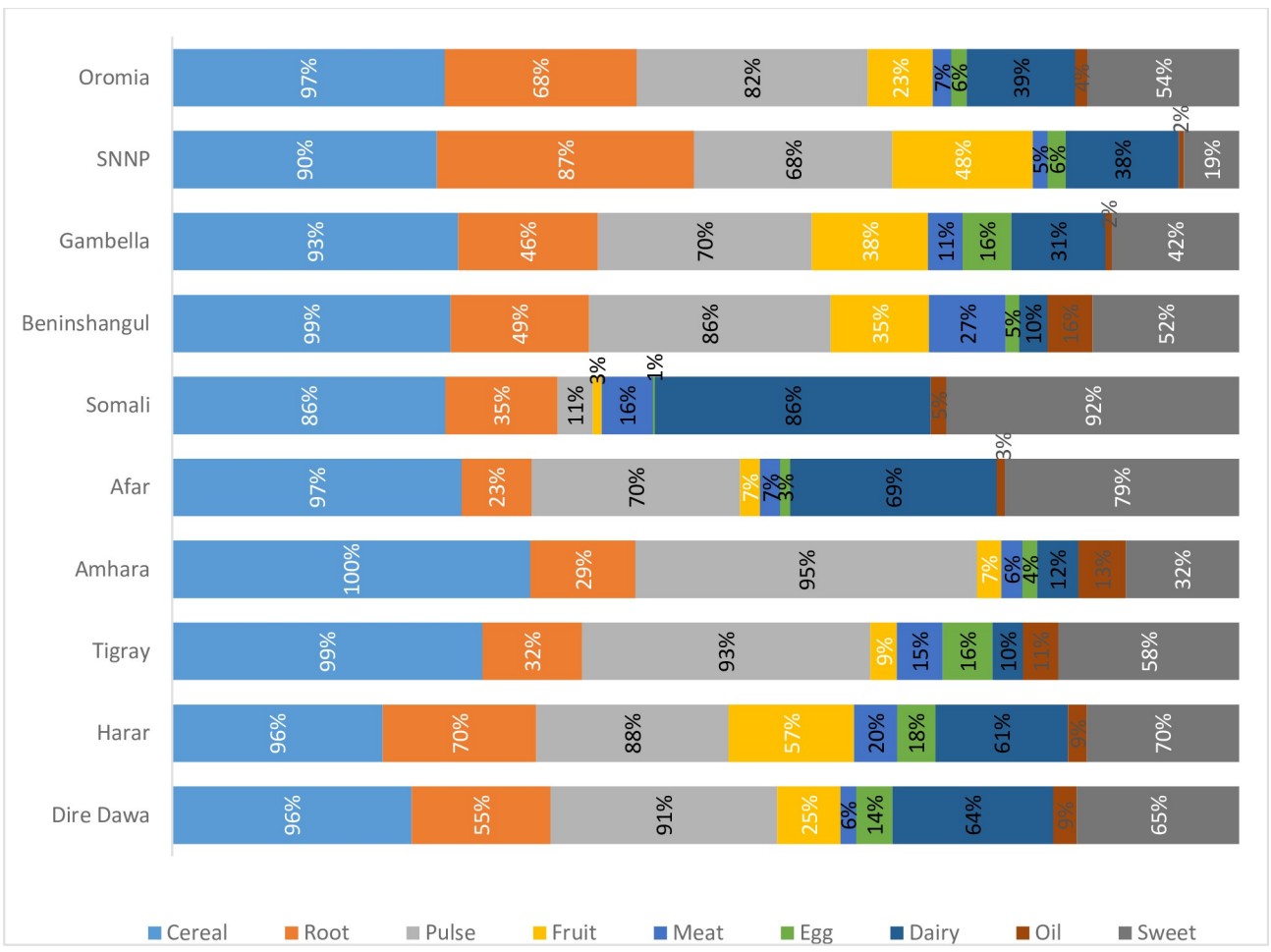

**Fig 1. Food groups consumed across regions in Ethiopia.**

## 3.3. Determinants of Household Dietary Diversity (HDD)

This part of the article focuses on the identification of socio-economic and demographic factors affecting household dietary diversity. *Sex of Household Head*: The finding of this study indicated that female headed households have 38 percent odds of consuming more diverse foods compared to male headed households, which is significant at p<0.01 (Table 3). This means that households headed by females have better dietary diversity than male-headed households do. They have a 4.99 mean dietary diversity value that falls under high dietary category compared male-headed households who have a dietary diversity value of 4.77. The finding of this study is consistent with the study of [24]. The possible explanation for this finding would be those households who are headed by females spend more on more diverse foods, and take the responsibilities of preparing different types of foods in the Ethiopian cultural context. Besides, women are in a better position in terms of familiarity with the nutritional benefits of different foods. They can, therefore, will have better information to make decision as to how to allocate family's budget on high quality foods [30].

**3.3.1. Education of household head.** The findings showed that education level of household head has significant correlation with consumption of diverse foods. This study showed that the odds of consuming diverse foods increases by 62 percent for those households who

**Table 3. Estimation results of the ordered logit model.**

|  | Odds Ratio | Std. Err | Z- Value | 95% CI |
|---|---|---|---|---|
| **Household head sex** |  |  |  |  |
| Male | Ref |  |  |  |
| Female | 1.38 | .16 | 2.81*** | 1.10; 1.73 |
| **Age** | 0.998 | 0.00 | -0.53 | .99; 1.00 |
| **Education** |  |  |  |  |
| No education | Ref |  |  |  |
| Primary | 1.35 | 0.14 | 3.00*** | 1.11; 1.64 |
| Secondary and above | 1.62 | 0.27 | 2.88*** | 1.2; 2.30 |
| **Marital status** |  |  |  |  |
| Currently married | Ref |  |  |  |
| Currently single | 0.63 | 0.078 | -3.73*** | 0.50; 0.80 |
| **Religion** |  |  |  |  |
| Orthodox | Ref |  |  |  |
| Muslim | 0.97 | 0.12 | -.022 | 0.76; 1.25 |
| Protestant and others | 1.18 | 0.17 | 1.2 | 0.90; 1.60 |
| **Household size** |  |  |  |  |
| 1–3 | Ref |  |  |  |
| 4–5 | 1.33 | 0.13 | 2.81*** | 1.09; 1.60 |
| 6 or more | 1.43 | 0.15 | 3.44*** | 1.17; 1.75 |
| **Region, Clustered** |  |  |  |  |
| Tigray and Amhara | Ref |  |  |  |
| Oromia and _SNNP | 2.05 | 0.26 | 5.59*** | 1.60,2.64 |
| BNG and Gambela | 1.96 | 0.30 | 4.34*** | 1.44,2.65 |
| Somali and _Afar | 2.58 | 0.42 | 5.86*** | 1.88,3.54 |
| Diredawa and Harare | 6.56 | 1.19 | 10.35*** | 4.60,9.37 |
| **Wealth Index** |  |  |  |  |
| Poorest | Ref |  |  |  |
| Poorer | 2.06 | 0.25 | 5.88*** | 1.62; 2.62 |
| Middle | 3.29 | 0.43 | 9.07*** | 2.54; 4.26 |
| Richer | 4.77 | 0.65 | 11.51*** | 3.65; 6.21 |
| Richest | 9.01 | 1.30 | 15.18*** | 6.79; 11.98 |
| **PSNP status** |  |  |  |  |
| Yes | Ref |  |  |  |
| No | 0.990 | 0.01 | -0.94 | 0.80; 1.21 |

*** *Significant at p<0.01*

**Significant *at p<0.05*

* *Significant at p<0.1*

have completed secondary and above level of education compared to those households headed by illiterates. Similarly, the odds of dietary diversity increases by 35 percent for those households who have completed primary education compared to those households headed by illiterates (Table 3). This indicates that when the educational status of household heads increase, the chance of consuming diverse foods will also increase. Our finding is in line with the findings of [8, 9, 13]. Evidence also showed that as the education status of rural household heads increase, their awareness about the importance of consuming diversified food increases [31].

**3.3.2. Marital status.** The result of the marital status as predictor of household dietary diversity found to be significant (P <0.01). Those households who are currently single have a

37 percent less chance of consuming diverse foods compared to those household heads who are currently married, implying that being married has positive implication for improved dietary diversity at household level (Table 3). The finding is consistent with [32]. The possible explanation can be those married households probably have responsibilities to diversify and increase incomes to access diverse foods.

**3.3.3. Household size.** The result indicated that household size is a significant predictor of consumption of diverse foods. This result indicated that households whose members are 6 and above have 43 percent chance of consuming diverse foods compared to households whose members are three and below. Similarly, households with four and five family members, their odds of dietary diversity increases by 33 percent compared to those households with three and below family members (Table 3). As size of families increase, the propensity to generate incomes outside farming increases, which also increases access to diverse food commodities. The find of this study is in line with the work of [8, 33].

**3.3.4. Location of household.** Regions were clustered based on the food consumption pattern as observed in Fig 1. Regional cluster was considered as a variable to capture the effect of locational factor on household dietary diversity. The analysis of the location variables is significant at p<0.01. Those households located in Harari regional state and in the rural surroundings of Diredawa town have 6.56 times more chance of consuming diverse foods compared to those households living in Tigray and Amhara regional states. Similarly, those households living in Afar and Somalia regional states have 2.6 times more chance of consuming diverse foods compared to households living in Tigray and Amhara regional states (Table 3).

**3.3.5. Wealth index.** The finding of this study indicated that there was a significant association between the wealth status of households and dietary diversity. The finding indicated that households who are in the upper wealth category have 9 times more chance of consuming diverse foods compared to those households who are the lower wealth category(P <0.01). In the same manner, Table 3 shows that the middle wealth group households have 3 times more chance of consuming diverse foods compared to the poorest group (P <0.01). This finding is in line with the study of [34–36]. It is intuitive that rich households have better purchasing power to access a variety of foods and also engage in the production of diversified food items.

## 4. Conclusions and policy implications

This study estimated the status and determinants of household dietary diversity in Ethiopia. The findings indicated that dietary diversity among rural households is low. Though livestock production is one of the dominant livelihood source for rural households in Ethiopia, consumption of livestock products is very low. The result further highlighted that household size, wealth status, location, education status, marital status and sex of heads of the household significantly determined dietary diversity in Ethiopia. The findings of this study are supported by Sen's entitlement thesis, the sustainable livelihood approach and UNICEF's nutrition analysis conceptual model. In this line, wealth status captures trade-based entitlement, which further improves access to diverse foods. Similarly, education and household size are factors that improves labour-based entitlement, which again improves access to diverse foods. The SLA and UNICEF's conceptual models also supported our findings of marital status and sex of household heads in determining dietary diversity in Ethiopia. More specifically, marital status and sex of household head are aspects of social capital and societal norms in which married families and female-headed households prepare and consume more diverse foods, *citrus paribus*, they have access to incomes. The location factor is also explained by UNICEF's analytical model. In this line, location factor captures differences in societal norms, which affects food

preferences and choices. On the contrary, participation in PSNP is found to have no significant effect on household's dietary diversity though the programme was designed to improve the livelihood of poor households in Ethiopia.

Based on the findings, it is recommended that the government and development partners need to take into considerations context specific interventions whenever they design and implement programmes that aim at improving household dietary diversity. This is to say, food consumption patterns were similar in Afar and Somali; Amhara and Tigray; Oromia and Southern nations and nationalities; and Benshangual Gumuz and Gambella regional states, and the level of dietary diversity among the cluster regions were significantly different. Hence, the study recommends the need for addressing similarities in food consumption patterns and differences in dietary diversity among these cluster regions while designing program interventions. In addition to encouraging farmers to diversify agricultural productions, it is also important to consider nutrition education for improved dietary diversity particularly consumption of livestock products. The finding calls for further in-depth study to investigate the relevance of PSNP interventions in view of improving household dietary diversity in Ethiopia.

## Acknowledgments

The authors used a publicly available dataset produced by the Central Statistical Agency of Ethiopia (now named as the Ethiopian Statistical Services) through the funding of the World Bank, Bill and Melinda Gates Foundation and the Foreign Commonwealth and Development Office. The title of the survey is the Ethiopia Socioeconomic Survey (ESS4) 2018–2019 with Reference No: ETH_2018_ESS_v01, and downloaded from: https://microdata.worldbank.org/index.php/catalog/3823#metadata-metadata_production [20] on 06 June 2022. The authors duly acknowledge the data producers and sponsors. However, omissions and/or errors in the article are the full responsibilities of the authors.

## Author Contributions

**Conceptualization:** Workicho Jateno.

**Data curation:** Workicho Jateno.

**Formal analysis:** Workicho Jateno.

**Methodology:** Workicho Jateno, Bamlaku Alamirew Alemu, Maru Shete.

**Writing – original draft:** Workicho Jateno.

**Writing – review & editing:** Workicho Jateno, Bamlaku Alamirew Alemu, Maru Shete.

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
