## [Decision Letter · Decision Letter 0]

28 Dec 2022

PONE-D-22-27221Rural household dietary diversity across regions in Ethiopia: Evidence from Ethiopian socio-economic survey dataPLOS ONE

Dear Dr. Jateno,

Thank you for submitting your manuscript to PLOS ONE. After careful consideration, we feel that it has merit but does not fully meet PLOS ONE’s publication criteria as it currently stands. Therefore, we invite you to submit a revised version of the manuscript that addresses the points raised during the review process.

We look forward to receiving your revised manuscript.

Kind regards,

Demisu Zenbaba Heyi, MPH

Academic Editor

PLOS ONE

Journal Requirements:

2. "PLOS requires an ORCID iD for the corresponding author in Editorial Manager on papers submitted after December 6th, 2016. Please ensure that you have an ORCID iD and that it is validated in Editorial Manager. To do this, go to ‘Update my Information’ (in the upper left-hand corner of the main menu), and click on the Fetch/Validate link next to the ORCID field. This will take you to the ORCID site and allow you to create a new iD or authenticate a pre-existing iD in Editorial Manager. Please see the following video for instructions on linking an ORCID iD to your Editorial Manager account: " ext-link-type="uri" xlink:type="simple">https://www.youtube.com/watch?v=_xcclfuvtxQ"

Reviewers' comments:

Reviewer's Responses to Questions

**Comments to the Author**

1. Is the manuscript technically sound, and do the data support the conclusions?

Reviewer #1: No

2. Has the statistical analysis been performed appropriately and rigorously? 

Reviewer #1: No

3. Have the authors made all data underlying the findings in their manuscript fully available?

Reviewer #1: Yes

4. Is the manuscript presented in an intelligible fashion and written in standard English?

Reviewer #1: Yes

5. Review Comments to the Author

Reviewer #1: Even if, this paper addressed important area of researchable topic with enough sample size, the following points should be addressed before publishing this paper.

Line1: Title of the study is that equivalent to household dietary diversity? If not what is there difference?

Line19: Result: better to start by answering your objectives. Additionally include AOR with 95% CI to the predictor variables.

Line 114: Variables: Which one is the validated tool in Ethiopia to measure household food security from the four methods you listed? And why you preferred HDDS? Again this part should be discussed in the introduction session.

Line 129: Table 1: I don’t understand the importance of this table. Better if you remove it.

Line 132: wealth index analysis method should be described under data analysis sub heading.

139-172: Data analysis method is not well explained at all. You need to write what you did in steps. No need of explaining the formula for the model you used.

Line196: Table 2: the title of the table should be self-explanatory

Line209: which statistical method you used to check the difference in HDDS across regions? It is not explained.

Line270 Table 4: see the above comment.

Line 247: Which model you applied to identify independent factors associated with HDDS? Have controlled confounders? Why you didn’t use different subheadings for result and discussion?

6. PLOS authors have the option to publish the peer review history of their article (what does this mean?). If published, this will include your full peer review and any attached files.

Reviewer #1: No

quillbot-extension-portal/quillbot-extension-portal

---

## [Author Response · Author response to Decision Letter 0]

2 Feb 2023

'Response to Reviewers'

Journal Requirements:

Response: We have attempted to follow and maintain PLOS ONE manuscript writing requirements 

"PLOS requires an ORCID iD for the corresponding author in Editorial Manager on papers submitted after December 6th, 2016. Please ensure that you have an ORCID iD and that it is validated in Editorial Manager. To do this, go to ‘Update my Information’ (in the upper left-hand corner of the main menu), and click on the Fetch/Validate link next to the ORCID field. This will take you to the ORCID site and allow you to create a new iD or authenticate a pre-existing iD in Editorial Manager. Please see the following video for instructions on linking an ORCID iD to your Editorial Manager account: https://www.youtube.com/watch?v=_xcclfuvtxQ"

Response: All the authors have ORCID ID 

Reviewers' comments:

Reviewer's Responses to Questions

Comments to the Author

1. Is the manuscript technically sound, and do the data support the conclusions?

Reviewer #1: No

Response: The conclusion section is revisited as per the reviewer’s suggestion (see the comment track in the conclusion section for the change). However, the study is conducted under a social science setting in which experimental design with controls is not feasible. The sample size for the study is big enough to represent the rural household population as we have used a data set from a national survey. The approach we followed is to control variables that explain dietary diversity of households by accounting for them through regression estimation so that we avoid omitted variable bias. The type of regression suitable for estimating is the ordered logit model which estimates the contributions of each variables (in our case confounding variables) that determine the probability of falling into a particular dietary diversity category. 

2. Has the statistical analysis been performed appropriately and rigorously?

Reviewer #1: No

Response: Based on this, we have done the following

• Abstract: Under result paragraph, we have included AOR with 95% CI based on the comments.

• Additional explanations are included under “variable and method of data analysis” sections (Refer to 2.2 and 2.3 sub headings). The explanations in the methodology section indicates that we have used appropriate and rigorous methods, analysis and interpretation as deemed appropriate as per the nature of the variables and objectives of the study. 

3. Have the authors made all data underlying the findings in their manuscript fully available?

Reviewer #1: Yes

4. Is the manuscript presented in an intelligible fashion and written in standard English?

Reviewer #1: Yes : 

 Additional: We have taken the leverage to further edit the language and improve the presentations of the technical findings so as to improve its technical soundness.

 

5. Review Comments to the Author

Reviewer #1: Even if this paper addressed important area of researchable topic with enough sample size, the following points should be addressed before publishing this paper.

Line1: Title of the study is that equivalent to household dietary diversity? If not what is there difference?

Response: The title is to refer to household dietary diversity. We were trying to be more specific by including ‘Rural’ in the title as the study considered only those households living in rural areas. As per the comment of the reviewer, we removed the term ‘rural’ in the title, and mentioned the focus of the study under ‘’the scope’’ section of the manuscript. 

Line19: Result: better to start by answering your objectives. Additionally include AOR with 95% CI to the predictor variables.

Response: As per the comment, AOR for each predictor variable is included in the result sections of the abstract

Line 114: Variables: Which one is the validated tool in Ethiopia to measure household food security from the four methods you listed? And why you preferred HDDS? Again this part should be discussed in the introduction session.

Response: Commonly food security studies in Ethiopia adopted individual calorie intake/ household caloric consumption to measure food security status and its determinants. In this case, the focus is on diet quality rather than total energy intake: The DDS takes into account the diversity of foods consumed, which can provide a more nuanced understanding of diet quality than the calorie consumption method, which only considers total energy intake. The measurement indicator (i.e dietary diversity) is also scientifically proved by various scholars as shown in the manuscript. See the response in the method section of manuscript .

Line 129: Table 1: I don’t understand the importance of this table. Better if you remove it.

Response: the comment is well taken. The table is replaced with texts.

Line 132: wealth index analysis method should be described under data analysis sub heading.

Response: Detail description of the procedure to construct wealth index is presented in the method section of the manuscript.

139-172: Data analysis method is not well explained at all. You need to write what you did in steps. No need of explaining the formula for the model you used.

Response: The section is elaborated based on the comments.

Line196: Table 2: the title of the table should be self-explanatory

Response: Modified as suggested

Line209: which statistical method you used to check the difference in HDDS across regions? It is not explained.

Response: Chi square analysis was done to show dietary diversity across regions. This is presented in the last column of table 2 of the original manuscript. We explained in the revised version the statistical tool we used to examine DDS across regions 

Line270 Table 4: see the above comment.

Response: We feel that removing table 4 (in the revised version, table 3) will reduce the technical soundness/scientific status of the manuscript, and suggest to maintain it.

Line 247: Which model you applied to identify independent factors associated with HDDS? Have controlled confounders? Why you didn’t use different subheadings for result and discussion?

Responses: 

• Which model you applied to identify independent factors associated with HDDS?

We applied the ordered logit model to identify independent factors associated with DDS. It is explained under the ‘Method’ section of the manuscript. 

• Have controlled confounders?

In the method section of the manuscript, the tools we adopted to identify/control confounding factors are explained. We used Variance Inflation Factor (VIF) for continuous and Contingency Coefficient (CC) for discrete variables to identify confounding variables, and if exist to exclude them from the ordered logit model.

• Why you didn’t use different subheadings for result and discussion?

We feel providing discussion with results in the same heading gives better opportunity to get information at the same place. It is a matter of convention/style. If the reviewer still feels that this is the best approach, we are willing to do it. But, we leave it under same sub-heading for now.

6. PLOS authors have the option to publish the peer review history of their article (what does this mean?). If published, this will include your full peer review and any attached files.

Do you want your identity to be public for this peer review? For information about this choice, including consent withdrawal, please see our Privacy Policy.

Reviewer #1: No

---

## [Editor Report · Decision Letter 1]

13 Feb 2023

PONE-D-22-27221R1Household dietary diversity across regions in Ethiopia: Evidence from Ethiopian socio-economic survey dataPLOS ONE

Dear Dr. Jateno,

Thank you for submitting your manuscript to PLOS ONE. After careful consideration, we feel that it has merit but does not fully meet PLOS ONE’s publication criteria as it currently stands. Therefore, we invite you to submit a revised version of the manuscript that addresses the points raised during the review process.

If applicable, we recommend that you deposit your laboratory protocols in protocols.io to enhance the reproducibility of your results. Protocols.io assigns your protocol its own identifier (DOI) so that it can be cited independently in the future. For instructions see: https://journals.plos.org/plosone/s/submission-guidelines#loc-laboratory-protocols. Additionally, PLOS ONE offers an option for publishing peer-reviewed Lab Protocol articles, which describe protocols hosted on protocols.io. Read more information on sharing protocols at https://plos.org/protocols?utm_medium=editorial-emailutm_source=authorlettersutm_campaign=protocols.

We look forward to receiving your revised manuscript.

Kind regards,

Demisu Zenbaba Heyi, MPH

Academic Editor

PLOS ONE

Additional Editor Comments:

data analysis method need major revision

Which model was used multinomial or ordinal logistic regression??

Assumption to use one of this model is not indicated???

Reviewers' comments:

quillbot-extension-portal/quillbot-extension-portal

---

## [Author Response · Author response to Decision Letter 1]

16 Feb 2023

1. Data analysis method need major revision: This section is revised including additional explanation on how data analysis were carried out

2. Which model was used multinomial or ordinal logistic regression??: Ordered logit model

3. Assumption to use one of this model is not indicated??? : explanation included (see section 2.4)

---

## [Editor Report · Decision Letter 2]

3 Mar 2023

PONE-D-22-27221R2Household dietary diversity across regions in Ethiopia: Evidence from Ethiopian socio-economic survey dataPLOS ONE

Dear Dr. Jateno,

Thank you for submitting your manuscript to PLOS ONE. After careful consideration, we feel that it has merit but does not fully meet PLOS ONE’s publication criteria as it currently stands. Therefore, we invite you to submit a revised version of the manuscript that addresses the points raised during the review process.

We look forward to receiving your revised manuscript.

Kind regards,

Demisu Zenbaba Heyi, MPH

Academic Editor

PLOS ONE

Additional Editor Comments:

In abstract 95% CI is not indicated??

In table 1 Mean(stand.dev/range) ...mean can be reported with standard deviation not with range.

 quillbot-extension-portal/quillbot-extension-portal

---

## [Author Response · Author response to Decision Letter 2]

6 Mar 2023

Comment: In abstract 95% CI is not indicated??

Response: In the abstract, the CI was mentioned/included without mentioning 95%. This time following the comment, 95% is added to explain more at what level the CI was generated.

Comment: In table 1 Mean (stand.dev/range) ...mean can be reported with standard deviation not with range.

Response: Range was deleted from the column title, and standard deviation was calculated and included for ‘’Household size’’ variable.

---

## [Editor Report · Decision Letter 3]

13 Mar 2023

Household dietary diversity across regions in Ethiopia: Evidence from Ethiopian socio-economic survey data

PONE-D-22-27221R3

Dear Dr. Jateno,

We’re pleased to inform you that your manuscript has been judged scientifically suitable for publication and will be formally accepted for publication once it meets all outstanding technical requirements.

Kind regards,

Demisu Zenbaba Heyi, MPH

Academic Editor

PLOS ONE

Additional Editor Comments (optional):

Reviewers' comments:

quillbot-extension-portal/quillbot-extension-portal

---

## [Editor Report · Acceptance letter]

28 Mar 2023

PONE-D-22-27221R3 

Household dietary diversity across regions in Ethiopia: Evidence from Ethiopian socio-economic survey data 

Dear Dr. Jateno:

I'm pleased to inform you that your manuscript has been deemed suitable for publication in PLOS ONE. Congratulations! Your manuscript is now with our production department. 

Kind regards, 

on behalf of

Dr. Demisu Zenbaba Heyi 

Academic Editor

PLOS ONE